# Evaluating Families’ Opinions of Routine Influenza Vaccination in Children Under 5 Years of Age in Spain

**DOI:** 10.3390/vaccines13010054

**Published:** 2025-01-10

**Authors:** Sílvia Burgaya-Subirana, Anna Ruiz-Comellas, Queralt Miró Catalina, Mònica Balaguer

**Affiliations:** 1Department of Paediatrics, Equip d’Atenció Primària de Manlleu, Institut Català de la Salut, Gerència d’Atenció Primària i a la Comunitat de la Catalunya Central, C/Castellot 17, 08560 Manlleu, Spain; sburgaya.cc.ics@gencat.cat; 2Faculty of Medicine, University of Vic-Central University of Catalonia (UVic-UCC), 08500 Vic, Spain; monica.balaguer@umedicina.cat; 3Department of Medicine, Equip d’Atenció Primària de Sant Joan de Vilatorrada, Institut Català de la Salut, Gerència d’Atenció Primària i a la Comunitat de la Catalunya Central, Avinguda del Torrent del Canigó, 0, 08250 Sant Joan de Vilatorrada, Spain; 4Unitat de Suport a la Recerca de la Catalunya Central, Fundació Institut Universitari per a la Recerca a l’Atenció Primària de Salut Jordi Gol i Gurina, 08272 Sant Fruitós de Bages, Spain; 5Unitat de Cures Intensives Pediàtriques, Institut de Recerca Sant Joan de Déu (IRSJD), Hospital Sant Joan de Déu, Santa Rosa 39-57, Esplugues de Llobregat, 08950 Barcelona, Spain

**Keywords:** influenza, influenza vaccination, vaccine hesitancy, knowledge, attitude, opinion

## Abstract

Background/Objectives: Influenza vaccination is the main method for preventing influenza. The objectives of this study are to evaluate the opinions of families on influenza vaccination and to determine the acceptance of influenza vaccination as a routine vaccine in children under 5 years of age. Methods: The method used was a descriptive cross-sectional study based on an ad-hoc survey. Between October 2023 and January 2024, an online survey was conducted among families with children between 6 months and 14 years of age attending paediatric consultations at a health centre. Results: A total of 388 families were surveyed. Out of these, 22.68% reported having ever vaccinated their children against influenza. The main reason for having them vaccinated was having received the recommendation from the paediatrician (68.18%). While 53.61% agreed with routine influenza vaccination, 53.09% did not intend to vaccinate their children against influenza in the 2023/24 period. The reasons for not vaccinating in 2023/24 were unawareness of the disease (29.41%), fear of unwanted effects of vaccination (27.94%) and lack of information about vaccination (19.61%). The reasons for vaccination in 2023/24 were protection of the child (81.87%), recommendation by the paediatrician (43.41%) and protection of the general population and susceptible persons (20.33%). Conclusions: Routine influenza vaccination is accepted by half of the parents. A lack of risk perception of the disease, concern about vaccine safety and lack of information are the main reasons for not vaccinating. It is essential to follow the health professionals’ recommendation to vaccinate.

## 1. Introduction

Seasonal influenza is a disease with a major public health impact [1]. The World Health Organization (WHO) estimates that 1 billion people are infected with influenza each year, 3–5 million of whom develop a severe form of the disease. The number of deaths from influenza-related respiratory illnesses is between 290,000 and 650,000 (case fatality rate: 0.1–0.2%) [2]. The child population is the most affected by this infection [3] and is the main transmitter of the virus [4]. In Spain, the highest cumulative incidence rate for influenza in the periods 2021/22 and 2022/23 was observed in the 0–4 age group, followed by the 5–14 age group [5,6]. Influenza hospitalisations in the same period were highest in the elderly, followed by the 0–4 years age group [5,6]. Therefore, children under 5 years of age are the most susceptible to influenza and the most vulnerable to severe infection [7].

Vaccination is considered one of the most important measures to prevent influenza or reduce the risk of severe cases, hospitalisations and deaths [3,8]. Some studies have even reported that paediatric influenza vaccination may not only protect children but also protect the entire community and reduce the incidence of influenza in the general population [8]. Despite all this, childhood influenza vaccination coverage in Spain is low [3,9]. A study conducted by our group observed an average influenza vaccination coverage of 28.8% between 2018 and 2023 [10].

Vaccine hesitancy in general is considered by the WHO as one of the top ten threats to global health [11] and was first defined in 2012 by the Strategic Advisory Group of Experts (SAGE) working group on vaccine hesitancy as the delayed acceptance or refusal of vaccination despite the availability of vaccination [12,13]. Vaccine hesitancy is complex and context-specific, such that it can vary according to time, place, and vaccine [12]. It may also vary depending on the perceived safety and effectiveness of the vaccine [11], as well as the perceived risk of the disease [14]. A study conducted in the United States in 2020 observed a 25.8% hesitancy towards influenza vaccination compared to a 6.1% hesitancy towards other routine vaccines [11]. Another study in the Netherlands observed that 59% of parents thought that influenza was not a serious enough disease to be vaccinated against, and only 15% intended to vaccinate their children against influenza [14].

Influenza vaccine hesitancy can be attributed to many causes: unawareness of the disease burden in children, concerns about the vaccine’s efficacy and/or safety, lack of knowledge about the seasonal vaccination program, false beliefs about influenza vaccination or lack of information in general [15,16].

In Spain, until the 2022/23 period, influenza vaccination in paediatrics was recommended for children older than 6 months with a high-risk disease for complications [17], but from the 2023/24 period and following the recommendations of the Spanish Ministry of Health and the WHO, the indications for influenza vaccination were expanded to include children aged 6 months to 5 years, as well as those older than 5 years of age with risk factors [18,19,20]. It is important to explain that in Spain, the flu vaccine is free for children belonging to these groups with an indication for vaccination.

This article is part of a line of research on childhood vaccination in Catalonia (Spain), a territory with a child population of 1,000,000 children. The first article published in “Vaccines” examined childhood influenza vaccination rates between 2018 and 2023 [10].

The aim of this study is to evaluate the opinions of families on influenza vaccination during the first systematic flu vaccination season in Catalonia and across Spain and to determine the acceptance of influenza vaccination as a routine vaccine in children under 5 years of age in paediatrics.

## 2. Materials and Methods

### 2.1. Study Design and Participants

A descriptive cross-sectional study was carried out based on an ad-hoc online questionnaire. Between October 2023 and January 2024, a survey was conducted among families with children between 6 months and 14 years of age who came to the Manlleu Primary Care Centre to have a consultation for any reason.

The Manlleu Primary Care Centre is part of public health program and is located in Central Catalonia (Spain). It serves a population of 21,000 inhabitants and 3500 children and is one of the largest in the region. It has a great diversity of population because of a high level of immigration (30%) in this area. Most of these come from Morocco and Sub-Saharan Africa. In the past five years, there have been 200 births per year. Regarding influenza vaccination, the centre administers 2500 doses annually (for both adults and children). Approximately 250 doses are administered to paediatric patients. This represents 25% of the influenza vaccination coverage.

Families or legal guardians with only one child under 6 months of age were excluded because the influenza vaccine cannot be administered in this group. Children over 14 years of age are not visited by a paediatrician in Spain; for this reason, they were also excluded from the study. Those with language barriers or comprehension difficulties, or those who did not want to answer the survey, were excluded too. 

The calculation of the sample was made with Granmo calculator, developed by de Gironí Heart Registry Program (REGICOR), IMIM, Barcelona, version 8.0. The link of the calculator is as follows: https://www.datarus.eu/aplicacines/granmo. Accessed on 21 May 2024. 

In order to estimate the sample calculation, the main objective of this study has been considered: the population estimate of the families’ opinion regarding vaccination and its acceptance. Not knowing the expected value of these proportions, a proportion of 0.5 was considered to be as conservative as possible, with an accuracy of 5% and confidence of 95%. Failure to follow up was not considered.

Thus, a minimum sample of 385 observations was required to ensure population estimates with a 95% confidence interval and a 5% precision level.

The sample was obtained using a convenience-based consecutive sampling method from the study population. A total of 390 families were invited to answer the survey. Out of this, 2 rejected participating because of lack of time.

### 2.2. Data Collection

Families or legal guardians attending the paediatric or paediatric nursing consultations at the Primary Care Centre for any reason were informed about the study and invited to participate voluntarily. It was explained that all data were anonymous and that the research team could not have access to individual participant data. Those who agreed to participate were provided with a QR code granting access to the online questionnaire via Microsoft 365 Forms, a corporate tool. Families could answer the questionnaire in the consultation room with the help of the health professional or at home. The health professional helped interpret the questionnaire for foreign families who spoke Castilian Spanish or Catalan well but had difficulties reading in both languages. Families with difficulties with the management of the mobile phone were helped by the health professional too.

### 2.3. Measurements

The questionnaire (Appendix A) was self-constructed and especially designed for this study. It was distributed in two languages: Castilian Spanish and Catalan (the two official languages in Catalonia). The questionnaire consisted of 19 questions pre-packaged by the researchers with a last open option. It was divided into 2 parts. The first part asked about socio-demographic data (age of parents, sex, country of origin, level of education, number of children and age of all children at paediatric age), chronic diseases of the children, influenza vaccination of the children in previous periods and reasons that the families had for vaccinating or not vaccinating their children in the past. The second part asked about the intention to vaccinate against influenza in the 2023/24 period, reasons for vaccinating and not vaccinating their children against influenza during this period, the opinions that families had about routine vaccination in children under 5 years of age, parents’ influenza vaccination in previous seasons and parents’ reasons for their vaccination or not. 

### 2.4. Statistical Analysis

Absolute frequencies and percentages were used to describe the characteristics of the sample participants and their opinions. The Chi-square test (*X*^2^) was used to analyse the relationship between variables.

Independent variables were as follows: age of parents, sex, country of origin, level of education, number of children and age of all children at paediatric age, chronic disease of children, influenza vaccination of the children in previous periods and influenza vaccination of parents in previous periods.

Dependent variables were intention to vaccinate in 2023/24 and reasons to vaccinate or not vaccinate.

Finally, to evaluate the magnitude of association between variables, the Odds Ratio (OR) was estimated by using the odds ratio function of the epitools package of the R software. Version 4.2.1. This is calculated through a 2 × 2 matrix, and the contrast used to obtain the *p*-value is the independence contrast. Confidence intervals were 95%; a significance level of 5% was set; and all analyses were performed with R statistical software version 4.2.1.

### 2.5. Ethical Considerations

The research was conducted in accordance with the Declaration of Helsinki and Spanish national and institutional legislation concerning clinical research and personal data protection. Participation in the study was voluntary and anonymous, guaranteeing absolute confidentiality. The study was conducted via an online survey, and as the data were anonymous, the signing of informed consent was not required. However, by completing the survey, participants were considered to have given their consent to participate in the study.

The study was approved by the IDIAP Jordi Gol Ethics Committee (code 23-/139). Date of approval: 26 July 2023).

## 3. Results

### 3.1. Characteristics of the Sample

A total of 388 families were surveyed. Almost 80% of the responses were answered by female mothers or legal guardians. The mean age of the parents who responded to the survey was 38.55 years (SD 7.29). Fifty-three percent were Spanish, and thirty-one percent were from North Africa. In terms of educational level, 33.51% of the parents had higher education. In the survey conducted, 30.33% of the paediatric-aged children of the families who responded were between 0 and 2 years old, and 29.33% were between 6 and 10 years old (Table 1). The majority of the children (84.54%) had no underlying disease. The majority, 72.42%, had never vaccinated their children against influenza, and 50.26% of the parents had never been vaccinated against influenza. Of the parents who have ever been vaccinated against the flu, the majority (81.33%) do not do so every year (Table 2).

### 3.2. Reasons for Having or Not Having Their Children Vaccinated Against Influenza in the Past

The main reasons parents stated for vaccinating their children in previous periods were as follows: “the paediatric professional recommended vaccination” (68.18%) and “my child has an underlying health condition” (37.5%). The main reason for not having previously vaccinated their children was that the health professional had not recommended vaccination (34.16%). The second most frequent reason (32.74%) was that the child had no underlying health condition. Other reasons were as follows: “influenza is not a serious illness for children” (11.7%), “I do not have enough information about the vaccine” (9.6%), “I am afraid of the unwanted effects of the vaccine” (8.9%), “the influenza vaccine is not effective” (5.3%), “in general, I do not trust vaccination” (4.6%) and “I have had a bad experience with influenza vaccination” (4.3%) (Table 3).

### 3.3. Reasons for Parents’ Vaccination

The reasons most parents gave for getting themselves vaccinated against influenza were as follows: “I am a person with a risk disease” (42.3%), “because I am a relative of a person with a risk disease” (23.1%) and “I am a healthcare worker” (19.2%) (Table 3).

### 3.4. Intention to Vaccinate in the 2023/24 Period: Reasons to Vaccinate and Not to Vaccinate

When asked about the intention to vaccinate their children against influenza in the 2023/24 period, the majority, 53.09%, responded that they did not intend to vaccinate their children, despite the fact that 53.61% agreed with systematic vaccination in children under 5 years of age. The reasons cited by parents for vaccinating their children during the 2023/24 period were as follows: “to protect my child” (81.87%), “because the paediatric professional recommended it” (43.41%) and “to protect the general population, especially the elderly and people with risk diseases, from influenza” (20.33%). Other reasons were “because I have enough information about the vaccine and I think it is necessary” (7.1%), “because other relatives or acquaintances have vaccinated their children and I trust their judgment” (3.8%) and “because I have had influenza in the past and know what the consequences may be” (3.8%).

The main reasons parents referred to for not vaccinating during the 2023/24 period were “influenza is not a serious disease” (29.41%), “I am afraid of unwanted effects of vaccination” (27.94%) and “I do not have enough information about vaccination” (19.61%). Less frequently, they answered, “the influenza vaccine is not effective” (11.8%), “I have had a bad experience with influenza vaccination” (11.3%), “the paediatric professional has not recommended it to me” (9.3%), “in general, I do not trust any vaccine” (9.3%), and “vaccination is painful and I feel bad that my child would have to be pricked with a needle” (2.9%) (Table 3). A significant percentage (28.92%) gave other reasons for not vaccinating (Table 4).

### 3.5. Variables Associated with Intention to Vaccinate During the 2023/24 Period

A significant relationship has been observed between the intention to vaccinate and the fact of having previously vaccinated the child, agreeing or disagreeing with vaccination in children under 5 years of age, and the parents having been vaccinated every year or ever for influenza (Table 5).

In terms of the magnitude of the association, it was observed that children with parents from the areas of Africa other than North Africa (OR:136.6 [3.82; 93.9]) or from the rest of Europe (OR:4.06 [1.14; 19.5]) were more likely to be vaccinated compared to the children of Spanish origin; children whose parents had ever vaccinated them against influenza had an OR: 4.03 [1.14; 19.5] compared to those who did not; children whose parents agreed with routine vaccination in children under 5 years had an OR:131 [39.6; 866]; children whose parents had ever been vaccinated against influenza or who did not remember had an OR:30.1 [1.96; 4.66] and 4.77 [2.03; 12.3], respectively, and children whose parents were vaccinated each year for influenza had an OR:5.68 [2.07; 20.05] (Table 6).

## 4. Discussion

Influenza vaccination is the main measure to prevent influenza [3,8], but its coverage in Spain is low despite the fact that vaccination is free in risk groups [3,9,10]. Following the recommendations of the WHO and the Spanish Ministry of Health, the 2023/24 period was the first in which all Spanish Autonomous Communities systematically vaccinated all children under 5 years of age [18,19,20]. In our study, we wanted to assess the acceptance and opinions of families regarding the new measures implemented and influenza vaccination in general. To date, this is the first study to describe the parents’ opinions of routine influenza vaccination during the first vaccination period across Spain.

Most of the surveys were answered by women. This is explained by the fact that mothers are the ones who usually accompany their children to the doctor and are the ones who were offered to answer the survey. The majority (72.42%) of respondents had never vaccinated their children against influenza. The main reason they gave for not having vaccinated their children was that the health professional had not informed them of the vaccination. Similarly, the main reason given by respondents who had previously vaccinated their children was recommendation by the paediatric professional. These results highlight the importance of paediatric health care professionals recommending influenza vaccination. Some studies have shown that their recommendation of influenza vaccination decreases vaccine hesitancy [8,21,22,23,24,25,26,27,28,29]. Health professionals should take every opportunity to recommend the influenza vaccination.

Regarding the intention to vaccinate their children against influenza in the 2023/24 period, the majority (53.09%) were not intending to vaccinate their children; low risk perception and concern about the safety of the influenza vaccine are the main barriers to influenza vaccination, as well as the lack of information. Previous studies corroborate this lack of information or misinformation and attribute it to the media, where false information is often reported [8,25,26,29,30]. Expanding knowledge about influenza vaccination and conducting vaccination campaigns promoted by health institutions, as well as carrying out awareness campaigns for health professionals to recommend flu vaccination, are some of the key measures to improve childhood influenza vaccination coverage. 

A total of 56.61% agreed with routine vaccination in children under 5 years of age. As for the association between vaccination intention and sociodemographic variables, statistical significance was only observed between Spanish children and children whose parents came from an area of Africa other than North Africa or from other European countries. These children are 13.6 and 4.03 times more likely, respectively, to be vaccinated against influenza than Spanish children. This is probably due to cultural reasons. Regarding the relationship between parental educational level and influenza vaccination, there are disparate results in the literature. Some studies report that the high educational level of parents increases the hesitancy to vaccinate [8,21], while others report that parents with a high educational level are more likely to vaccinate against influenza [11,25]. Our study has not shown statistical significance between educational level and intention to vaccinate. This may be because the information and awareness of influenza in children in Spain remain low across all educational groups. Finally, we have observed that having previously vaccinated the children and the parents previously having been vaccinated is related to a higher intention to vaccinate in the 2023/24 period. Awad S et al. observed the same, that 87.4% of parents who had never been vaccinated against influenza had not vaccinated their children against influenza [28].

The main limitation of this study was that it was conducted in only one primary care centre. This may lead to selection bias, as the characteristics of the participants may not be representative of the general population. Patients from a single, specific centre may share certain socioeconomic, geographic, or demographic factors that differentiate them from the rest of the population. This could affect the generalisability of the study results and their extrapolation to other clinical settings with differing conditions. As a result, external validity could be compromised. Nonetheless, we chose to conduct the study at the Manlleu Primary Care Centre because it is one of the largest in the region, and additional locations were unlikely to reveal different patterns or results.

Another limitation was the inability to include families with a language barrier. Although these constitute a very small proportion of the population, their exclusion could contribute to selection bias.

The questionnaire required responses predefined by the researchers. While this method facilitates correction, it may also introduce the limitation of suggesting answers based on social desirability criteria. However, during the design of the questionnaire, this method was deemed preferable as it ensured consistency in responses, saved time, facilitated data analysis, reduced ambiguities, and improved the efficiency of the preparation process. This approach also minimised the risk of incomplete responses and reduced subjectivity in their interpretation. To mitigate this limitation, the final option for each question allowed participants to respond openly. Another potential issue with the questionnaire was self-reporting bias. Participants may have deliberately provided misleading answers.

## 5. Conclusions

Having received a recommendation for vaccination from a healthcare professional is the leading cause of influenza vaccination in the paediatric age group, which is why both paediatricians and paediatric nurses should be encouraged to recommend influenza vaccination to families. In the coming years, training campaigns for health professionals should be carried out with the aim of increasing influenza vaccination coverage in the paediatric age group.

## Figures and Tables

**Table 1 vaccines-13-00054-t001:** Description of the sample (sociodemographic data).

Variable	Category	Absolute Frequency (*n* = 388)	Percentage (%)
Sex	Female	308	79.38
	Male	80	20.62
Country of origin:	Central America	6	1.55
	Rest of Europe	12	3.09
	South America	20	5.15
	Rest of Africa	23	5.93
	North Africa	121	31.19
	Spain	206	53.09
Age of parents (*n* = 379)	20–30 years old	40	10.55
	30–40 years old	172	45.38
	40–50 years old	141	37.20
	50–60 years old	26	6.86
Educational level:	Primary	73	18.81
	Secondary	92	23.71
	Further Education	93	23.97
	Higher Education	130	33.51
Number of children:	1	128	32.99
	2	166	42.78
	3	58	14.95
	4	25	6.44
	5	6	1.55
	6	5	1.29
Age range of children (*n* = 633)	6 months-2 years old	192	30.33
	3–5 years old	141	22.27
	6–10 years old	185	29.23
	11–14 years old	115	18.17

**Table 2 vaccines-13-00054-t002:** Description of the variables of interest.

Variable	Category	Absolute Frequency (*n* = 388)	Percentage (%)
Do your children have any illnesses?	No	328	84.54
	Yes	60	15.46
Have you ever vaccinated your child against influenza?	No	281	72.42
	Yes	88	22.68
	I do not know	19	4.90
If vaccination is indicated, do you intend to vaccinate your child against influenza this year?	No	206	53.09
	Yes	182	46.91
Do you agree that all children under 5 years of age should be vaccinated against influenza?	No	112	28.87
	Yes	208	56.61
	I have no opinion	68	17.53
Have you ever had an influenza vaccination?	No	195	50.26
	Yes	166	42.78
	I do not know	27	6.96
Do you get an influenza vaccination every year?	No	135	81.33
	Yes	31	18.67

**Table 3 vaccines-13-00054-t003:** Description of the reasons for having or not having their children vaccinated in previous periods and in the 2023/24 period.

	Percentage (%)
**Why did you vaccinate your children? (*n* = 88)**	
Paediatric recommendation	68.2
Child with risk disease	37.5
Others	5.7
Person with risk disease at home	3.4
**What was the reason for not vaccinating your children in past seasons? (*n* = 281)**	
No paediatric recommendation	34.2
No disease	32.7
Other	20.6
Underestimating influenza	11.7
No information	9.6
Fear	8.9
Ineffectiveness	5.3
Distrust	4.6
Bad experience	4.3
**What are your reasons for vaccinating your child this season? (*n* = 182)**	
To protect the child	81.9
Paediatric **recommendation**	43.4
To protect the population	20.3
Sufficient information	7.1
Family/acquaintances criteria	3.8
Influenza consequences	3.8
Other	0.5
**What are your reasons for not vaccinating your child this season? (*n* = 204)**	
Underestimating influenza	29.4
Other	28.9
Fear	27.9
No information	19.6
Ineffectiveness	11.8
Bad experience	11.3
No paediatric recommendation	9.3
Distrust	9.3
Vaccine pain	2.9
**What is the reason for your vaccination? (*n* = 26)**	
I have a risk disease.	42.3
Family member with risk disease	23.1
I am a healthcare worker.	19.2
Other	15.4

**Table 4 vaccines-13-00054-t004:** Other reasons given by families for not vaccinating their children in the 2023/24 period.

Children must be immunised naturally.
It is not an essential vaccine.
If the child does not have any illness, there is no need to vaccinate him/her against influenza.
Children are too young for influenza vaccination.
My son has never had influenza.
Too many vaccines are given these days.
The vaccine does not guarantee protection against being infected with influenza.
It is a vaccine for older people.
It is a new vaccine.
Lack of time.
Currently my child does not get sick.
We are awaiting surgery and do not want our child to get a fever in case they call us.
It is not necessary to vaccinate every year for influenza.

**Table 5 vaccines-13-00054-t005:** Bivariate table. Association between intention to vaccinate and sociodemographic variables and variables of interest.

	Do You Intend to Vaccinate Your Child Against Influenza this Year?		
**Variable**	**No** **(*n* = 206)**	**Yes** ***n* = 182**	** *p* ** **-Value**
**Sex:**			0.133
Female	170 (82.5%)	138 (75.8%)	
Male	36 (17.5%)	44 (24.2%)	
**Country of origin:**			**<0.001**
Central America	2 (0.97%)	4 (2.20%)	
Spain	120 (58.3%)	86 (47.3%)	
North Africa	72 (35.0%)	49 (26.9%)	
Rest of Africa	2 (0.97%)	21 (11.5%)	
Rest of Europe	3 (1.46%)	9 (4.95%)	
South America	7 (3.40%)	13 (7.14%)	
**Age of parents: (No *n* = 202; Yes *n* = 177)**			0.379
20–30 years old	22 (10.9%)	18 (10.2%)	
30–40 years old	90 (44.6%)	82 (46.3%)	
40–50 years old	80 (39.6%)	61 (34.5%)	
50–60 years old	10 (4.95%)	16 (9.04%)	
**Educational level:**			0.198
Further Education	52 (25.2%)	41 (22.5%)	
Primary	32 (15.5%)	41 (22.5%)	
Secondary	46 (22.3%)	46 (25.3%)	
Higher Education	76 (36.9%)	54 (29.7%)	
**Number of children:**			0.602
1	62 (30.1%)	66 (36.3%)	
2	95 (46.1%)	71 (39.0%)	
3	31 (15.0%)	27 (14.8%)	
4	14 (6.80%)	11 (6.04%)	
5	2 (0.97%)	4 (2.20%)	
6	2 (0.97%)	3 (1.65%)	
**Do your children have any illnesses?**			0.920
No	175 (85.0%)	153 (84.1%)	
Yes	31 (15.0%)	29 (15.9%)	
**Have you ever vaccinated your child against influenza?**			**<0.001**
No	181 (87.9%)	100 (54.9%)	
I do not know.	7 (3.40%)	12 (6.59%)	
Yes	18 (8.74%)	70 (38.5%)	
**Do you agree that all children under 5 years of age should be vaccinated against influenza?**			**<0.001**
No	110 (53.4%)	2 (1.10%)	
I have no opinion.	38 (18.4%)	30 (16.5%)	
Yes	58 (28.2%)	150 (82.4%)	
**Have you ever had an influenza vaccination?**			**<0.001**
No	131 (63.6%)	64 (35.2%)	
I do not know.	8 (3.88%)	19 (10.4%)	
Yes	67 (32.5%)	99 (54.4%)	
**Do you get an influenza vaccination every year?**			**0.001**
No	63 (94.0%)	72 (72.7%)	
Yes	4 (5.97%)	27 (27.3%)	
	**No *n* = 346**	**Yes *n* = 287**	** *p* ** **-value**
**Children’s ages:**			0.225
6 months-2 years old	93 (26.9%)	99 (34.5%)	
11–14 years old	67 (19.4%)	48 (16.7%)	
3–5 years old	80 (23.1%)	61 (21.3%)	
6–10 years old	106 (30.6%)	79 (27.5%)	

**Table 6 vaccines-13-00054-t006:** Magnitude of the association between intention to vaccinate and the different variables.

Variable	OR	*p*-Value
**Sex:**		
Female	Ref.	Ref.
Male	1.50 [0.92; 2.48]	0.106
**Country of origin:**		
Central America	2.68 [0.48; 22.1]	0.263
Spain	Ref.	Ref.
North Africa	0.95 [0.60; 1.50]	0.827
Rest of Africa	13.6 [3.82; 93.9]	**<0.001**
Rest of Europe	4.03 [1.14; 19.5]	**0.030**
South America	2.56 [0.99; 7.16]	0.052
**Age of parents:**		
20–30 years old	Ref.	Ref.
30–40 years old	1.11 [0.56; 2.25]	0.765
40–50 years old	0.93 [0.46; 1.91]	0.844
50–60 years old	1.93 [0.70; 5.47]	0.202
**Educational level:**		
Further Education	Ref.	Ref.
Primary	1.62 [0.87; 3.03]	0.127
Secondary	1.27 [0.71; 2.27]	0.425
Higher Education	0.90 [0.53; 1.55]	0.706
**Number of children:**		
1	Ref.	Ref.
2	0.70 [0.44; 1.12]	0.137
3	0.82 [0.44; 1.53]	0.532
4	0.74 [0.30; 1.77]	0.500
5	1.81 [0.32; 15.0]	0.510
6	1.37 [0.20; 12.2]	0.743
**Do your children have any illnesses?**		
No	Ref.	Ref.
Yes	1.07 [0.61; 1.86]	0.810
**Have you ever vaccinated your children against influenza?**		
No	Ref.	Ref.
I do not know.	3.06 [1.18; 8.60]	**0.021**
Yes	6.96 [4.00; 12.7]	**<0.001**
**Do you agree that all children under 5 years of age should be vaccinated against influenza?**		
No	Ref.	Ref.
I have no opinion.	39.8 [11.2; 274]	**<0.001**
Yes	131 [39.6; 866]	**0.000**
**Have you ever had an influenza vaccination?**		
No	Ref.	Ref.
I do not know.	4.77 [2.03; 12.3]	**<0.001**
Yes	3.01 [1.96; 4.66]	**<0.001**
**Do you get an influenza vaccination every year?**		
No	Ref.	Ref.
Yes	5.68 [2.07; 20.5]	**<0.001**
	**OR**	** *p* ** **-value**
**Children’s ages:**		
6 months-2 years old	Ref.	Ref.
11–14 years old	0.67 [0.42; 1.07]	0.097
3–5 years old	0.72 [0.46; 1.11]	0.136
6–10 years old	0.70 [0.47; 1.05]	0.086

## Data Availability

The data that support the findings of this study are available from the corresponding author upon request.

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
