# Peer review of "Evaluating Families’ Opinions of Routine Influenza Vaccination in Children Under 5 Years of Age in Spain"

_vaccines, 2025, doi:10.3390/vaccines13010054_

Round 1

Reviewer 1 Report

Comments and Suggestions for Authors

1. Why was the study conducted in only one location? Do you think including additional locations might have revealed different patterns or results?

2. Why did this study fail to find a statistically significant correlation between parents' educational level and their intention to vaccinate?

Author Response

Reviewer 1

Comments: 

  1. Why was the study conducted in only one location? Do you think including additional locations might have revealed different patterns or results?

  1. Why did this study fail to find a statistically significant correlation between parents' educational level and their intention to vaccinate?

Response: 

  1. The study was conducted at a single location because the calculated required sample size was 385. The Primary Health Care Centre where the study was conducted is one of the largest in the region, allowing for the collection of the entire sample at this site. In addition, It is a Primary Care Centre with a great diversity of population. It has 30% immigration (especially people of Moroccan origin and Sub-Saharan Africa).  We believe that including additional locations would not have revealed different patterns or results. This has been explained in greater detail in the paper. 

  1. We believe this study failed to find a statistically significant correlation between parents' educational level and their intention to vaccinate, possibly because information and awareness of influenza in children in Spain remains low across all educational groups. (LInes 280-281).

Reviewer 2 Report

Comments and Suggestions for Authors

The manuscript by Burgaya-Subirana and coworkers describes the results from a questionnaire of families visiting a pediatric clinic on their attitudes toward influenza vaccination. Families coming for any reason to the clinic for consultation were asked to participate. The important, in my opinion, results are (1) health professionals should explain the need for vaccination, and that might increase uptake; (2) North Africans and Africans coming from outside this region have a very different attitude toward vaccination.

The limitations of the study are more than just taking participants from one primary care centre and excluding people with language barrier. Having information from one centre is a major problem if the area it attracts patients from is very biased. Can you tell us a bit more about the centre and the potential difference between the reached populace and the populace of Catalonia? Furthermore, what about those who do not go to doctors at all? Do you have some statistics on the portion of the population with children who is basically invisible to the health care system? Also, what bias could stem from the reason patients visit the health care centre? For example, if a good portion of them visit because of influenza-like symptoms then those falling ill because not being vaccinated would be overrepresented if it is influenza season (I know most children had no illness according to their caregiver).

I think that irrespective of how much the sample is representative of Catalonia and/or Spain, the main message that doctors should communicate the importance of influenza vaccine more stands. But some of the other significant results might be because of the biased sample.

I suggest the manuscript to be accepted with minor changes.

Minor comments

Page3Line102: The Χ2 should contain a superscript 2.

P4L132 There is a period before [Figure 1], also Figure 1 should be surrounded by normal parantheses.

There are several issues with Figure 1. The titles of the subpanels are but below the panel, whereas I would expect them above. The title mostly overlaps in vertical space with the X axis label (Percentage) of the next panel. Texts are generally too small. But to be honest, this Figure should not be a figure but a table. That would be much easier to read.

P9L208: I think the reference to Awad et al. should not be Awad S et al.

Author Response

Reviewer 2

Comments: 

The manuscript by Burgaya-Subirana and coworkers describes the results from a questionnaire of families visiting a pediatric clinic on their attitudes toward influenza vaccination. Families coming for any reason to the clinic for consultation were asked to participate. The important, in my opinion, results are (1) health professionals should explain the need for vaccination, and that might increase uptake; (2) North Africans and Africans coming from outside this region have a very different attitude toward vaccination.

The limitations of the study are more than just taking participants from one primary care centre and excluding people with language barrier. Having information from one centre is a major problem if the area it attracts patients from is very biased. Can you tell us a bit more about the centre and the potential difference between the reached populace and the populace of Catalonia? Furthermore, what about those who do not go to doctors at all? Do you have some statistics on the portion of the population with children who is basically invisible to the health care system? Also, what bias could stem from the reason patients visit the health care centre? For example, if a good portion of them visit because of influenza-like symptoms then those falling ill because not being vaccinated would be overrepresented if it is influenza season (I know most children had no illness according to their caregiver).

I think that irrespective of how much the sample is representative of Catalonia and/or Spain, the main message that doctors should communicate the importance of influenza vaccine more stands. But some of the other significant results might be because of the biased sample.

I suggest the manuscript to be accepted with minor changes.

Minor comments

Page3Line102: The Χ2 should contain a superscript 2.

P4L132 There is a period before [Figure 1], also Figure 1 should be surrounded by normal parantheses.

There are several issues with Figure 1. The titles of the subpanels are but below the panel, whereas I would expect them above. The title mostly overlaps in vertical space with the X axis label (Percentage) of the next panel. Texts are generally too small. But to be honest, this Figure should not be a figure but a table. That would be much easier to read.

P9L208: I think the reference to Awad et al. should not be Awad S et al.

Response:

"The centre where we conducted the survey is part of public health and is located in Central Catalonia. It serves a population of 21,000 inhabitants, including approximately 3,500 children, and is one of the largest in the region. The key difference between the surveyed population and the population of Catalonia is the socio-economic level, as most immigrants have a lower socio-economic status than Spanish nationals.

In Spain, the National Health Service is public, and children are one of the most protected groups. Whether or not they have a health card, children must be seen if they visit a medical centre. As a result, we have not identified any population that does not visit the doctor. All residents in our region are covered by the health system, and there is no population invisible to healthcare services."

Page 3 Line 102: A superscript has been changed (X2 is X2). (Now is line 145)

P4L132: The figure 1 has been changed to a table (table 3). 

P9L208:  The reference Awad S et al is correct. In this study they say textually: “A total of 848 (87.4%) of 970 people who never received the influenza vaccine themselves had also not vaccinated their children against the flu.” The text can be found in page 3306 in the point 3.5. titled: “Factors related to the attitudes and practices of parents concerning childhood influenza vaccination”. (Now line 280). We referenced the citation as it appears in PubMed."Awad S, Abdo N, Yusef D, Jawarneh A, Babaa A, Alwady D, Al-Bawayeh H, Kasrawi I, Amaireh M, Obeidat M, Bany Amer N, Alonze S. Knowledge, attitudes and practices related to influenza illness and vaccination in children: Role of awareness campaigns in changing parents' attitudes toward influenza vaccination in Jordan. Vaccine. 2019 May 31;37(25):3303-3309. doi: 10.1016/j.vaccine.2019.04.083. Epub 2019 May 6. PMID: 31072734.

Reviewer 3 Report

Comments and Suggestions for Authors

Thank you for the opportunity to review the manuscript ID: vaccines-3311162. This manuscript aimed to evaluate families' opinions of routine influenza vaccination in children under 5 years of age in Spain.

There are numerous inconsistencies in the Methods and Results sections in this paper.      

Lines 80-81: Explain the rationale why families with children between 6 months and 14 years of age were included in this study. 

Lines 83-84: Specify the lower and upper limits for the ages of children that were suitable for this study. 

Lines 99-100: Based on what criteria was the sample size estimated? Cite the appropriate reference. 

Lines 103-104: Specify which test was applied to estimate the OR. 

On Table 1, is the category `0-2 years old` correctly written for the variable `Age range of children`? Or should it be written `6 months-2 years old`? 

Explain the rationale for the comparison according to the `Children's ages` variable for the presented 4 age categories. 

Explain how the results presented in Tables 4 and 5 are in accordance with the aim of this paper (I quote: `to evaluate the opinions of families on influenza vaccination and to determine the acceptance of influenza vaccination as a routine vaccine in children under 5 years of age in paediatrics.`). 

Lines 210-212: Limitations of this study are very sparsely stated and unsatisfactorily discussed.   

Author Response

Reviewer 3

Comments: 

Thank you for the opportunity to review the manuscript ID: vaccines-3311162. This manuscript aimed to evaluate families' opinions of routine influenza vaccination in children under 5 years of age in Spain.

 There are numerous inconsistencies in the Methods and Results sections in this paper.   

 Lines 80-81: Explain the rationale why families with children between 6 months and 14 years of age were included in this study. 

 Lines 83-84: Specify the lower and upper limits for the ages of children that were suitable for this study. 

 Lines 99-100: Based on what criteria was the sample size estimated? Cite the appropriate reference. 

 Lines 103-104: Specify which test was applied to estimate the OR. 

 On Table 1, is the category `0-2 years old` correctly written for the variable `Age range of children`? Or should it be written `6 months-2 years old`? 

 Explain the rationale for the comparison according to the `Children's ages` variable for the presented 4 age categories. 

Explain how the results presented in Tables 4 and 5 are in accordance with the aim of this paper (I quote: `to evaluate the opinions of families on influenza vaccination and to determine the acceptance of influenza vaccination as a routine vaccine in children under 5 years of age in paediatrics.`). 

Lines 210-212: Limitations of this study are very sparsely stated and unsatisfactorily discussed.

Response:

Line 80-81: We included families with children aged between 6 months and 14 years because the influenza vaccine can be administered from 6 months, and in Spain, all children are seen by a paediatrician up to the age of 14. The text has been revised to reflect this: 'Families with children under 6 months of age were excluded because the influenza vaccine cannot be administered to this group. Children over 14 years of age are not seen by a paediatrician in Spain, and therefore, they were also excluded from the study.' This has been explained in lines 97-101.

Line 83-84: The lower and upper limits for the ages of children that were suitable for this study were 6 months and 14 years. 6 months because, as It has been said before, influenza vaccine can only be administered in children more than 6 months and 14 years of age because in Spain the pediatricians visit the children up to 14 years old. It has been explained in lines 97-101. 

Line 99-100: The calculation of the sample was made with Granmo calculator, developed by de Gironí Heart Registry Program (REGICOR), IMIM, Barcelona, version 8.0. The link of the calculatro si: https://www.datarus.eu/aplicacines/granmo. 

In order to estimate the sample calculation the main objective of the study has been considered: the population estimate of the families’ opinion regarding vaccination and its acceptance. Not knowing the expected value of these proportions, a proportion of 0.5 was considered to be as conservative as possible, accuracy of 5% and confidence of 95%. Loss to follow-up was not considered. 

Thus, a minimum sample of 385 observations was required to ensure population estimates with a 95% confidence interval and a 5% precision level. 

Line 103-104: The odds ratio has been estimated with the odds ratio function of the epitools package of the R software. This is calculated through a 2X2 matrix and the contrast used to obtain the p-value is the independence contrast. 

On table 1 the category for children aged 0 to 2 years is correct. This information refers to the age of the respondents' children. To participate in the study, it was necessary to have at least one child aged between 6 months and 14 years. It is important to mention that respondents may have multiple children, and some of them may fall within the age range of 0 to 2 years. For example, a parent with an 8-year-old child and a 3-month-old baby would select both the '6–10 years' and '0–2 years' categories.

The rationale for the comparison based on the 'Children's ages' variable for the four presented age categories is as follows: four age categories were used because parents may have differing perceptions about influenza vaccination depending on their children's ages. Parents with younger children may be more concerned about influenza and thus more inclined to vaccinate, whereas parents with older children may perceive the effects of influenza as less severe and have less intention to vaccinate.

.

Tables 4 and 5 (now renumbered as Tables 5 and 6 due to another reviewer suggesting that Figure 1 be converted into a table) address the aim: 'Acceptance of influenza vaccination as a routine vaccine in children under 5 years of age in paediatrics.' The variable 'Do you agree that all children under 5 years of age should be vaccinated against influenza?' directly addresses this aim. Respondents who answered 'yes' showed greater acceptance of influenza vaccination. Similarly, individuals who answered 'yes' to the question 'Do you intend to vaccinate your child against influenza this year?' also demonstrated higher acceptance of the vaccine. The other variables in these tables are significant for understanding the characteristics of individuals who accept influenza vaccination.

The other aim, 'To evaluate the opinions of families on influenza vaccination,' is reflected in Figure 1 (now Table 3) and Table 3 (now Table 4). This has been further explained in the discussion section.

Line 210-212: The text has been changed: “The main limitation of the study was that it was conducted in only one primary care centre. This may lead to selection bias, as the characteristics of the participants may not be representative of the general population. Patients from a single, specific centre may share certain socioeconomic, geographic, or demographic factors that differentiate them from the rest of the population. This could affect the generalisability of the study results and their extrapolation to other clinical settings with differing conditions. As a result, external validity could be compromised.

Nonetheless, we chose to conduct the study at the Manlleu Primary Care Centre because it is one of the largest in the region, and additional locations were unlikely to reveal different patterns or results. Another limitation was the inability to include families with a language barrier. Although these constitute a very small proportion of the population, their exclusion could contribute to selection bias.

The questionnaire required responses predefined by the researchers. While this method facilitates correction, it may also introduce the limitation of suggesting answers based on social desirability criteria. However, during the design of the questionnaire, this method was deemed preferable as it ensured consistency in responses, saved time, facilitated data analysis, reduced ambiguities, and improved the efficiency of the preparation process. This approach also minimised the risk of incomplete responses and reduced subjectivity in their interpretation.

To mitigate this limitation, the final option for each question allowed participants to respond openly. Another potential issue with the questionnaire was self-reporting bias. Participants may have deliberately provided misleading answers.” (Now lines 286-307). 

Reviewer 4 Report

Comments and Suggestions for Authors

The authors discussed the topic of childhood influenza vaccination which has the advantage of protecting children and reducing the spread of infection in the general population. Vaccine hesitancy is high in Spain. The authors investigated parents' opinions in the 2023-24 season

1.     A piece of information that stands out is whether the flu vaccination is free in Spain. This is an important element and should be discussed.

2.     Limitations: The research only involved Catalan or Castilian speakers. Catalonia is a multi-ethnic reality and the share of people coming from other languages is not negligible. Even in the sample they are prevalent. Are the authors sure that they understood the questions?

3.     The consultation method (online, with QR code) requires good familiarity with the uses of the mobile phone. Many parents had only elementary education. Online research could have selected the sample. How many parents were invited to participate?

4.     At line 212 the authors mention possible selection bias, but they have made no attempt to measure the extent of this bias. The fact that the survey was conducted in a single center, could allow a measure. How many vaccines has the center administered? How many people live in the catchment area of ​​the center? How many births have there been in the previous five years? This information is necessary to know if this study has valid epidemiological foundations.

5.     The questionnaire required answers pre-packaged by the researchers. This method favors correction but may have the defect of suggesting answers by criteria of social desirability. An in-person interview would have allowed authors to detect any other answers. The authors should discuss this limitation.

Author Response

Reviewer 4

Comments: 

The authors discussed the topic of childhood influenza vaccination which has the advantage of protecting children and reducing the spread of infection in the general population. Vaccine hesitancy is high in Spain. The authors investigated parents' opinions in the 2023-24 season

  1.     A piece of information that stands out is whether the flu vaccination is free in Spain. This is an important element and should be discussed.

  1.     Limitations: The research only involved Catalan or Castilian speakers. Catalonia is a multi-ethnic reality and the share of people coming from other languages is not negligible. Even in the sample they are prevalent. Are the authors sure that they understood the questions?

  1.     The consultation method (online, with QR code) requires good familiarity with the uses of the mobile phone. Many parents had only elementary education. Online research could have selected the sample. How many parents were invited to participate?

  1.     At line 212 the authors mention possible selection bias, but they have made no attempt to measure the extent of this bias. The fact that the survey was conducted in a single center, could allow a measure. How many vaccines has the center administered? How many people live in the catchment area of the center? How many births have there been in the previous five years? This information is necessary to know if this study has valid epidemiological foundations.

  1.     The questionnaire required answers pre-packaged by the researchers. This method favors correction but may have the defect of suggesting answers by criteria of social desirability. An in-person interview would have allowed authors to detect any other answers. The authors should discuss this limitation.

Response:

  1. This Text has been added in line 75-77: “It is important to explain that in Spain, the flu vaccine is free for children belonging to the groups indicated for vaccination”. It has been mentioned in the discussion (line 241). 

  1. It is certain that Catalonia is a multi-ethnic region, and the proportion of people from different linguistic backgrounds is not negligible; however, the majority understand Castilian Spanish. If any language barriers were detected, the survey was not offered to those individuals. If the issue was that they had difficulty reading Castilian Spanish or Catalan but could understand the language, a healthcare professional assisted them in understanding the questions. This has been explained in lines 121-124 of the revised manuscript.

  1. Nowadays, individuals with elementary education are accustomed to using mobile phones and new technologies, particularly parents under 50 years of age. In any case, if anyone expressed difficulty using their device but wished to complete the survey, the health professional assisted them. We invited 390 parents to participate in the survey, of whom 2 declined. The reason for the refusal was not related to any technological barrier. It was due to lack of time. 

The following text has been added to lines 123-124 of the revised manuscript: 'Families with difficulties in managing the mobile phone were also assisted by the health professional.

  1. It has been changed text (line 90-96): “The Manlleu Primary Care Centre is part of public health and is located in Central Catalonia (Spain). It serves a population of 21,000 inhabitants and 3500 children and is one of the largest of the region. It has a great diversity of population because in this area there are a high level of immigration (30%). Most of these come from Morocco and Sub-Saharan Africa. In the past five years there have been 200 birth per year. Regarding influenza vaccination, the center administers 2,500 annually (for both adults and children). Approximetly 250 doses are administered to paediatric patients. This represents 25% the of influenza vaccination coverage.” 

  1. It has been added the following text in line 298: “The questionnaire required responses predefined by the researchers. While this method facilitates correction, it may also introduce the limitation of suggesting answers based on social desirability criteria. However, during the design of the questionnaire, this method was deemed preferable as it ensured consistency in responses, saved time, facilitated data analysis, reduced ambiguities, and improved the efficiency of the preparation process. This approach also minimised the risk of incomplete responses and reduced subjectivity in their interpretation. To mitigate this limitation, the final option for each question allowed participants to respond openly.”

Reviewer 5 Report

Comments and Suggestions for Authors

This study aims to evaluate the opinions of families on influenza vaccination and to determine the acceptance of influenza vaccination as a routine vaccine in children under 5 years of age in Spain. Some suggestions are provided for the authors to consider for the improvement of the manuscript.

General issue:

-            Please clarify the age of children included in the study. Both age under 5 years and age between 6 months and 14 years were mentioned.

Abstract:

-            Inconsistent study results: Recommendation from the paediatrician with 68.18% was reported as one of the main reasons for having the children vaccinated on Page 1 lines 24-25 but 43.41% was reported on line 29. Please clarify.

Introduction:

-            More work should be done on the literature about parental hesitancy to vaccinate their children against influenza. There should be review studies available, please cite and discuss.

-            Is there any theory used to guide the design of the study/measurements? If so, please provide more information. If not, please justify.

Methods:

-            Please provide inclusion and exclusion criteria for the participants.

-            Please create a subsection “Data collection” and provide relevant information.

-            Please create a subsection “Measurements” and provide details of the measurements. For instance, regarding the items about reasons for vaccination or not to vaccinate, were they self-constructed or adapted from previous studies? Please provide sample items and report reliability and validity when appropriate.

-            Please mention what the independent and dependent variables were in the “Statistical analysis”.

Results:

-            When reporting results of OR, please use complete paragraphs instead of point form.

Discussion:

-            More discussions about the practical implications regarding study results are expected.

-            There are other limitations of the study, such as issues related to generalization of the study results, self-report bias, and causal relationships between independent and dependent variables.

Conclusion:

-            Please write the conclusion concisely in one paragraph.

Look forward to reading the revised version of the manuscript.

Author Response

Reviewer 5

Comments: 

This study aims to evaluate the opinions of families on influenza vaccination and to determine the acceptance of influenza vaccination as a routine vaccine in children under 5 years of age in Spain. Some suggestions are provided for the authors to consider for the improvement of the manuscript.

General issue:

-            Please clarify the age of children included in the study. Both age under 5 years and age between 6 months and 14 years were mentioned.

Abstract:

-            Inconsistent study results: Recommendation from the paediatrician with 68.18% was reported as one of the main reasons for having the children vaccinated on Page 1 lines 24-25 but 43.41% was reported on line 29. Please clarify.

Introduction:

-            More work should be done on the literature about parental hesitancy to vaccinate their children against influenza. There should be review studies available, please cite and discuss.

-            Is there any theory used to guide the design of the study/measurements? If so, please provide more information. If not, please justify.

Methods:

-            Please provide inclusion and exclusion criteria for the participants.

-            Please create a subsection “Data collection” and provide relevant information.

-            Please create a subsection “Measurements” and provide details of the measurements. For instance, regarding the items about reasons for vaccination or not to vaccinate, were they self-constructed or adapted from previous studies? Please provide sample items and report reliability and validity when appropriate.

-            Please mention what the independent and dependent variables were in the “Statistical analysis”.

Results:

-            When reporting results of OR, please use complete paragraphs instead of point form.

Discussion:

-            More discussions about the practical implications regarding study results are expected.

-            There are other limitations of the study, such as issues related to generalization of the study results, self-report bias, and causal relationships between independent and dependent variables.

Conclusion:

-            Please write the conclusion concisely in one paragraph.

Look forward to reading the revised version of the manuscript.

Response:

General issue: 

  • As we mentioned in line 89, in the study we included families with children aged between 6 months and 14 years. We aimed to assess  their acceptance of the systematic vaccination in children under 5 years, which was a new measure implemented during the 2023/24 season across  Spain. 

Abstract: 

  • The percentage of 68.18% corresponds to individuals who vaccinated their children before the 2023/24 season and reported doing so based on a recommendation from their paediatrician. The percentage of 43.41% corresponds to individuals who intend to vaccinate during the 2023/24 season, also citing their paediatrician's recommendation as the reason. These two figures differ in terms of the timing of vaccination: one refers to vaccinations already administered, while the other refers to the intention to vaccinate. 

Introduction: 

  • It has been added the following text in line 66: “Influenza vaccine hesitancy can be attributed to many causes: unawareness of the disease burden in children, concerns about the vaccine efficacy and/or safety, lack of knowledge about the seasonal vaccination program, false beliefs about influenza vaccination or lack of information in general [16,17]”.
  • The study an the ad-hoc survey have been designed for the new incorporation of systematic influenza vaccination in children under 5 years old across Spain because we wanted to know the families’ opinion about this new measure. 

Methods: 

  • Inclusion criteria: families with children between 6 months and 14 years who went to the Manlleu Primary Care Centre for any reason. 
  • Exclusion criteria: families with only one child under 6 months, families with barriers language o families who do not wanted to answer the survey. 

All of this is explained between lines 88-99. 

  • A subsection “Data collection” has been created. (Line 116): “Families or legal guardians attending the pediatric or pediatric nursing consultations at the Primary Care Centre, for any reason, were informed about the study and invited to participate voluntarily. It was explained that all data were anonymous and that the research team could not have access to individual participant data. Those who agreed to participate were provided with a QR code granting access to the online questionnaire via Microsoft 365 Forms, a corporate tool. Families could answer the questionnaire in the consultation room with the help of the health professional or at home. The health professional helped interpret the questionnaire for foreign families who spoke Castilian Spanish or Catalan well but had difficulties reading in both languages. Families with difficulties with the management of the mobile phone were helped by the health professional too”.

  • A subsection “Measurments” has been created: “The questionnaire [Supplement 1] was self-constructed and especially designed for this study. It was distributed in two languages: Castilian Spanish and Catalan (the two official languages in Catalonia). The questionnaire consisted of 19 questions pre-packaged by the researchers with a last open option. It was divided into 2 parts. The first part asked about socio-demographic data (age of parents, sex, country of origin, level of education, number of children and age of all children at paediatric age), chronic diseases of the children, influenza vaccination of the children in previous periods and reasons that the families had had for vaccinating or not vaccinating their children in the past. The second part asked about the intention to vaccinate against influenza in the 2023/24 period, reasons for vaccinating and not vaccinating their children against influenza during this period, the opinions that families had about routine vaccination in children under 5 years of age and, parents' influenza vaccination in previous seasons and parents’ reasons for their vaccination or not”. 

  • This text has been added in “Statistical analysis” (line 147): “Independent variables were: age of parents, sex, country of origin, level of education, number of children an age of all children at paediatric age, chronic disease of children, influenza vaccination of the children in previous periods and influenza vaccination of parents in previous periods.Dependent variables were intention to vaccination in 2023/24 and reasons to vaccinate or not vaccinate”. 

Results: 

The reporting results of OR has been changed into paragraph. (Line 229-237): “In terms of the magnitude of the association, it was observed that those more likely to be vaccinated were: children with parents from areas of Africa other than North Africa OR:136.6 [3.82;93.9] or from the rest of Europe OR:4.06 [1.14;19.5] compared with children of Spanish origin; children whose parents had ever vaccinated them against influenza OR: 4.03 [1.14;19.5] compared to those who did not; children whose parents agreed with routine vaccination in children under 5 years OR:131 [39,6;866]; children whose parents had ever been vaccinated against influenza or who did not remember OR:30.1 [1.96;4.66] and 4.77 [2.03;12.3] respectively and children whose parents were vaccinated each year for influenza OR:5.68 [2.07;20.05]. [Table 5].”

Discussion

  • The discussion and limitations has been changed : 

“Influenza vaccination is the main measure to prevent influenza [3,8], but its coverage in Spain is low despite the fact that vaccination is free in risk groups [3,9,10]. Following the recommendations of the WHO and the Spanish Ministry of Health, the 2023/24 period was the first in which all Spanish Autonomous Communities systematically vaccinated all children under 5 years of age [19-21]. In our study we wanted to assess the acceptance and opinions of families regarding the new measure implemented and of influenza vaccination in general. To date, this is the first study to describe parents’ opinions of routine influenza vaccination during the first vaccination period across Spain.

Most of the surveys were answered by women. This is explained by the fact that mothers are the ones who usually accompany their children to the doctor and are the ones who were offered to answer the survey. 72.42% of respondents had never vaccinated their children against influenza. The main reason they gave for not having vaccinated their children was that the health professional had not informed them of the vaccination. Similarly, the main reason given by respondents who had previously vaccinated their children was recommendation by the paediatric professional. These results highlight the importance of paediatric health care professionals recommending influenza vaccination. Some studies have shown that their recommendation of influenza vaccination decreases vaccine hesitancy [22-31]. Health professionals should take every opportunity to recommend influenza vaccination.

Regarding the intention to vaccinate children against influenza in the 2023/24 period, the majority (53.09%) not intending to vaccinate their children, low risk perception and concern about the safety of the influenza vaccine are the main barriers to influenza vaccination, as well as lack of information. Previous studies corroborate this lack or misinformation and attribute it to the media, where false information is often reported [8,27,28,31,32]. Expanding knowledge about influenza vaccination and conducting vaccination campaigns promoted by health institutions are some of the key measures to improve childhood influenza vaccination coverage. As well as carrying out awareness campaigns for health professionals to recommend flu vaccination.

56.61% agreed with routine vaccination in children under 5 years of age. As for the association between vaccination intention and sociodemographic variables, statistical significance was only observed between Spanish children and children whose parents came from an area of Africa other than North Africa or from other European countries. These children are 13.6 and 4.03 times more likely, respectively, to be vaccinated against influenza than Spanish children. This is probably due to cultural reasons. Regarding the relationship between parental educational level and influenza vaccination, there are disparate results in the literature. Some studies report that the high educational level of parents increases the hesitancy to vaccinate [22,23], while others report that parents with a high educational level are more likely to vaccinate against influenza [14,27]. Our study has not shown statistical significance between educational level and intention to vaccinate. This may be due because information and awareness of influenza in children in Spain remains low across all educational groups. Finally, we have observed that having previously vaccinated the children and the parents previously having been vaccinated is related to a higher intention to vaccinate in the 2023/24 period. Awad S et al. observed the same, that 87.4% of parents who had never been vaccinated against influenza had not vaccinated their children against influenza [30].

The main limitation of the study was that it was conducted in only one primary care centre. This may lead to selection bias, as the characteristics of the participants may not be representative of the general population. Patients from a single, specific centre may share certain socioeconomic, geographic, or demographic factors that differentiate them from the rest of the population. This could affect the generalisability of the study results and their extrapolation to other clinical settings with differing conditions. As a result, external validity could be compromised. Nonetheless, we chose to conduct the study at the Manlleu Primary Care Centre because it is one of the largest in the region, and additional locations were unlikely to reveal different patterns or results.

Another limitation was the inability to include families with a language barrier. Although these constitute a very small proportion of the population, their exclusion could contribute to selection bias.

The questionnaire required responses predefined by the researchers. While this method facilitates correction, it may also introduce the limitation of suggesting answers based on social desirability criteria. However, during the design of the questionnaire, this method was deemed preferable as it ensured consistency in responses, saved time, facilitated data analysis, reduced ambiguities, and improved the efficiency of the preparation process. This approach also minimised the risk of incomplete responses and reduced subjectivity in their interpretation. To mitigate this limitation, the final option for each question allowed participants to respond openly. Another potential issue with the questionnaire was self-reporting bias. Participants may have deliberately provided misleading answers.”

Conclusion: 

  • Conclusions has been written concisely in only one paragraph: 

“Having received a recommendation for vaccination from a healthcare professional is the leading cause of influenza vaccination in the paediatric age group, which is why both paediatricians and paediatric nurses should be encouraged to recommend influenza vaccination to families. In the coming years, training campaigns for health professionals should be carried out with the aim of increasing influenza vacciantion coverage in the paediatric age group.”

Round 2

Reviewer 3 Report

Comments and Suggestions for Authors

Unfortunately, the main inconsistencies in this paper (primarily in the Methods and Results sections) remain in the revised version of this paper.

The biggest disadvantages:

Explain how the statements in the following sentences align:

Lines 20-22: `Between October 2023 and January 2024, an online survey was conducted among families with children between 6 months and 14 years of age attending pediatric consultations at a health center.`.

Lines 69-74: `In Spain, until the 2022/23 period, influenza vaccination in paediatrics was recommended for children older than 6 months with a high-risk disease for complications [18], but from the 2023/24 period and following the recommendations of the Spanish Ministry of Health and the WHO, the indications for influenza vaccination were extended to all children under 5 years of age and to those older than 5 years of age with risk factors [19-21].`.

Lines 97-98: `Families or legal guardians with only one child under 6 months of age were excluded because the influenza vaccine can not be administered in this group.`.   

And finally, as a consequence, the age categories of children presented in the Results section are not in agreement with the stated criteria for the inclusion of respondents (that is, families) in this paper. 

Author Response

Comment:

Unfortunately, the main inconsistencies in this paper (primarily in the Methods and Results sections) remain in the revised version of this paper.

The biggest disadvantages:

Explain how the statements in the following sentences align:

Lines 20-22: `Between October 2023 and January 2024, an online survey was conducted among families with children between 6 months and 14 years of age attending pediatric consultations at a health center.`.

Lines 69-74: `In Spain, until the 2022/23 period, influenza vaccination in paediatrics was recommended for children older than 6 months with a high-risk disease for complications [18], but from the 2023/24 period and following the recommendations of the Spanish Ministry of Health and the WHO, the indications for influenza vaccination were extended to all children under 5 years of age and to those older than 5 years of age with risk factors [19-21].`.

Lines 97-98: `Families or legal guardians with only one child under 6 months of age were excluded because the influenza vaccine can not be administered in this group.  

And finally, as a consequence, the age categories of children presented in the Results section are not in agreement with the stated criteria for the inclusion of respondents (that is, families) in this paper. 

Response:

We are pleased to answer your request.

As we explained in lines 69-74, in Spain, until the 2022/23 season, influenza vaccination in paediatrics was recommended for children older than 6 months with a high-risk disease for complications, but from the 2023/24 season and following the recommendations of the Spanish Ministry of Health and the WHO, the indications for influenza vaccination were extended to all children aged between 6 months to 5 years of age and to those older than 5 years of age with risk factors.

To clarify this point, it is important to note that until the end of March 2023, the influenza vaccination criteria for the paediatric age group in Spain were as follows: children older than 6 months with a high-risk disease for complications. From 15 October 2023, with the commencement of the new flu season, the criteria were updated to include children aged 6 months to 5 years and those older than 5 years with risk factors.

We included families with at least one child between 6 months to 14 years old because they were the group who could be vaccinated with influenza vaccine. (Before 6 months the vaccine can not be administered). Parents with only one child under 6 months were excluded because of the age of the children they had not previously considered the vaccination and this could be a limitation of the study and lead to erroneous conclusions. The first part of the survey asks about past vaccination and these families would not have been able to answer anything.

The “ages categories” in the results section is the answer to the following question in the survey:Within what age range are each of your children of paediatric age? (You can choose more than one option)”:

___ 6 months-2 years

___ 3-5 years

___ 6-10 years

___ 11-14 years

A typographical error has been detected in tables 1,5 and 6. In the age range of children it would have to be written 6 months-2 years instead of 0-2 years. We sincerely apologize for this error, which has now been corrected.

A sentence has been changed in the manuscript (line 72-74) so that is more understandable: “the indications for influenza vaccination were expanded to include children aged 6 months to 5 years, as well as those older than 5 years of age with risk factors [19-21].”

Reviewer 4 Report

Comments and Suggestions for Authors

The authors addressed all my comments

Author Response

Thank you for your comment. 

Reviewer 5 Report

Comments and Suggestions for Authors

Thanks for addressing the comments. 

Author Response

Thank you for your comment.